# Histogenesis of Atypical Teratoid Rhabdoid Tumors: Anatomical and Embryological Perspectives

**DOI:** 10.3390/cancers18010008

**Published:** 2025-12-19

**Authors:** Tadanori Tomita

**Affiliations:** 1Department of Neurological Surgery, Northwestern University Feinberg School of Medicine, Chicago, IL 60611, USA; tad.tomita@gmail.com; 2Ann & Robert H. Lurie Children’s Hospital of Chicago, 225 E. Chicago Avenue, Chicago, IL 60611, USA

**Keywords:** ATRT, brain tumor, children, neural crest, neural stem cell, SMARCB1, superior medullary velum

## Abstract

Atypical teratoid/rhabdoid tumors (ATRTs) are rare, highly aggressive CNS neoplasms in infancy and early childhood that can arise anywhere along the neuraxis and have extracranial rhabdoid counterparts. Converging genetic and developmental evidence implicates neural stem cells or neural crest-derived progenitors as candidate cells of origin, with tumorigenesis initiated by embryonic or early fetal loss of the SMARCB1 tumor suppressor. In an anatomical topographic analysis of a single-institution cohort of 50 pediatric patients with ATRT, integrated with an extensive literature review, molecular subclassification (ATRT-TYR, ATRT-SHH, ATRT-MYC) aligns with distinct anatomic niches and age distributions, suggesting subgroup-specific developmental contexts. The characteristic cellular heterogeneity—rhabdoid cells admixed with mesenchymal and/or epithelial elements—supports a progenitor at or near the neural plate border with partial neural crest competency, despite the neural crest’s little or no contribution to brain parenchyma. Definitively resolving ATRT histogenesis will require integrative, spatially informed transcriptomic and epigenomic analyses across CNS and extracranial rhabdoid tumors, coupled with developmental lineage modeling and age-stratified clinical data. Such efforts should clarify the timing and trajectory of SMARCB1 loss, define the permissive developmental windows and niches for transformation, and reveal subgroup-specific vulnerabilities to improve diagnosis, risk stratification, and potential targeted therapy.

## 1. Introduction

Beckwith and Palmer (1978) examined 427 archival Wilms tumor specimens and identified 8 cases (fewer than 2%) with distinctive histologic features characterized by sheets of polygonal cells with acidophilic cytoplasm and prominent vesicular nuclei [1]. These cells—later termed “rhabdoid” cells—often contained eosinophilic cytoplasmic inclusions composed of intermediate filaments, imparting their characteristic morphology; in some cases, the cells aggregated into epithelioid or alveolar configurations. At the time, these tumors were described as exhibiting a rhabdomyosarcomatoid pattern, suggesting myoblastic differentiation.

Haas et al. (1981) subsequently coined the term “rhabdoid tumor of the kidney” to describe this entity, highlighting its histologic resemblance to rhabdomyoblasts despite the absence of true skeletal muscle differentiation [2]. Immunohistochemical studies later demonstrated loss of INI1 (SMARCB1) expression, which has become a defining diagnostic hallmark of rhabdoid tumors. These neoplasms have since been reported at numerous anatomic sites, most commonly in the kidney [3]. Their neoplastic cells exhibit characteristic cytologic features—eccentric nuclei with prominent nucleoli and dense eosinophilic cytoplasm with paranuclear inclusions of intermediate filaments—correlating with an aggressive clinical course. The primary central nervous system counterpart was first described by Biggs et al. in 1987 as “malignant rhabdoid tumor of the CNS,” [4].

Malignant rhabdoid tumors (MRTs) can arise at virtually any anatomic site; those originating in the kidney are termed rhabdoid tumors of the kidney (RTKs). Historically, CNS rhabdoid tumors were grouped with primitive neuroectodermal tumors (PNETs) and were noted for divergent differentiation along neuronal, mesenchymal, and epithelial lineages. In 1996, Rorke et al. introduced the term “atypical teratoid/rhabdoid tumor” (ATRT) to reflect this heterogeneity [5]. The descriptor “rhabdoid” denotes the morphological resemblance to rhabdomyoblasts (immature skeletal muscle cells), whereas “teratoid” refers to the presence of diverse tissue elements reminiscent of teratomas derived from multiple germ layers.

ATRTs were formally recognized in the 2000 World Health Organization (WHO) Classification of Tumors of the Central Nervous System as embryonal neoplasms. They account for approximately 1.6% of all pediatric CNS tumors and exhibit a marked predilection for very young children: about 90% occur in patients younger than three years, with roughly 40–50% diagnosed within the first year of life [6,7,8,9].

On rare occasions, ATRT arises in the context of familial rhabdoid tumor predisposition syndrome (RTPS), which is associated with an even earlier age at presentation (typically 4–7 months) [10,11]. In one study, 66% of infants diagnosed within 28 days of birth harbored germline SMARCB1 mutations [12].

ATRTs can arise throughout the CNS, with most tumors occurring in the intracranial compartments; primary spinal presentations are relatively uncommon [5,13,14,15,16]. Classified as WHO grade IV, ATRTs are clinically aggressive, and despite multimodal therapeutic advances, outcomes remain suboptimal. In the Children’s Oncology Group trial ACNS0333, 4-year event-free survival (EFS) and overall survival (OS) were 37% and 43%, respectively, with younger age at diagnosis strongly associated with poorer prognosis [14]. The European Rhabdoid Registry (EU-RHAB) also reported similar results, a 5-year OS of 34.7 ± 4.5% and a 5-year EFS of 30.5 ± 4.2% [17,18]. The Japan Children’s Cancer Group observed 2-year progression-free survival (PFS) and OS rates of 66.6 ± 8.3% and 45.9 ± 8.7%, respectively, and 5-year rates of 44.2 ± 9.9% and 34.2 ± 8.9% [11]. Trials from St. Jude Children’s Research Hospital (SJYC07 and SJMB03) showed more favorable outcomes in average-risk patients (5-year PFS 73%, OS 82%), whereas outcomes in high-risk patients—defined by metastatic disease or residual tumor > 1.5 cm^2^—were substantially poorer (5-year PFS and OS 18%) [19].

Rhabdoid cells are unequivocally neoplastic and do not represent a normal somatic precursor population. They have no physiological counterpart at any stage of CNS development. The absence of a defined physiological progenitor that corresponds to “rhabdoid cells” complicates assignment of a cell of origin and hinders distinction from other primary CNS neoplasms on histology alone. Further complexity arises from histologically indistinguishable MRTs that occur in extracranial organs. Contemporary diagnosis of ATRT relies on demonstrating biallelic inactivation of SMARCB1 and less commonly, other SWI/SNF complex genes with malignant transformation driven by disruption of chromatin remodeling. Current hypotheses propose that ATRTs may arise from transformed embryonic stem cells, neural progenitors, and/or neural crest-related progenitors; however, the timing of oncogenic initiation and the definitive cell(s) of origin remain unresolved.

This communication delineates clinical features of ATRT from a neurosurgical perspective and synthesizes the literature on cytogenesis and tumorigenesis through anatomical and embryological lenses.

## 2. Clinical Materials and Methods

Over a 20-year period (2001–2020), the author managed 50 infants and children with atypical teratoid/rhabdoid tumor (ATRT) at Ann Robert H. Lurie Children’s Hospital of Chicago, Chicago, IL USA in the capacity of attending neurosurgeon and/or consultant. The cohort included patients from birth to 20 years of age with pathologically confirmed ATRT based on surgical tissue samples. All patients underwent open surgery with the intent of achieving gross total resection.

Diagnosis of ATRT was established by neuropathologists using histopathologic evaluation and immunohistochemical (IHC) studies. Histological characteristics included the presence of rhabdoid cells together with mixtures of small round cells and regions with neuroepithelial cells, epithelial and/or mesenchymal tissue. IHC showed demonstration of loss of nuclear INI-1 expression, a critical diagnostic marker for ATRT. Additional immunostains applied were synaptophysin, S-100 protein, and GFAP to assess neuroectodermal components, as well as cytokeratin, vimentin, epithelial membrane antigen, and smooth muscle actin to identify mesenchymal elements.

A retrospective review was conducted to characterize clinical presentations with emphasis on tumor location and structural involvement, integrating pre- and postoperative magnetic resonance imaging (MRI) and computed tomography (CT) with intraoperative observations abstracted from operative reports. The study was approved by the Institutional Review Board of Lurie Children’s Hospital (IRB#2005–12692) on 1 September 2005.

## 3. Demographics and Overall Results

The cohort comprised 27 males and 23 females. The age at diagnosis ranged from 1 to 134 months (mean, 28.9 months; median, 21 months), except for one 20-year-old woman with cerebellopontine angle (CPA) ATRT. Of these, 23 patients (46%) were diagnosed within the first year of life. Tumor location was infratentorial in 18 patients (36%), supratentorial in 15 (30%), pineal in 11 (22%), and spinal in 6 (12%). All patients underwent surgical resection with the intent to achieve gross-total resection. Preoperative neuroimaging, operative reports, and postoperative imaging were reviewed for each case. Pathologic analysis confirmed ATRT in all patients, with loss of SMARCB1/INI1 demonstrated in every case.

### 3.1. Supratentorial AT/RT

Among the 15 patients with supratentorial tumors. Patient age at diagnosis ranged from 3 to 47 months (mean, 25 months; median, 27 months).

Tumor epicenters and patterns of extension were best delineated by operative observations combined with preoperative and postoperative imaging (Figure 1). Three patients had tumors involving both the temporal and frontal lobes that crossed the Sylvian fissure (Figure 2). Despite appearance of thalamic involvement on preoperative imaging, two patients had tumors primarily occurred in the basal ganglia, which became more evident on postoperative MRI (Figure 3). Additional hemispheric tumor extended into the thalamus. These lesions were resected via a trans-sylvian approach.

Three tumors were predominantly located within the ventricle, two within the lateral ventricle and another within the third ventricle (Figure 4). None appeared to arise from the choroid plexus or as pedunculated lesions from the ependymal layer; rather, operative and imaging features supported a paraventricular origin because of diffuse attachment to the ventricular wall. Two large hemorrhagic hemispheric tumors extended into the lateral ventricle. These paraventricular ATRT extends into the lateral ventricle (Figure 5).

Three patients presented with massive subdural/parenchymal masses involving both cerebral hemispheres across the falx cerebri; two bi-frontal and one in bi-occipital (Figure 6). These tumors appeared to originate within the cerebral hemispheres and permeate or traverse the falx cerebri, resulting in bilateral disease.

Of the supratentorial tumors, 8 were solid and 7 demonstrated peritumoral cysts. Five showed signs of central necrosis, and 2 had intratumoral hemorrhage with associated intraventricular hemorrhage. Additional hemorrhagic foci were identified on susceptibility-weighted or gradient-echo sequences in others. Post-contrast MR imaging demonstrated heterogeneous enhancement following contrast infusion with variable degrees. Diffusion-weighted imaging uniformly showed restricted diffusion.

### 3.2. Infratentorial ATRT

Eighteen patients had tumors in the posterior fossa. The mean age at diagnosis was 29 months (range, 7 weeks to 20 years), and the median age was 10 months. Ten patients presented with obstructive hydrocephalus at presentation.

Five patients had tumors centered within the fourth ventricle; in four of these, the tumor involved the floor of the ventricle (brainstem) (Figure 7).

In five additional patients, tumors straddled both the fourth ventricle, the lateral recess and in some further the CPA and/or the cerebellomedullary fissure (CMF), producing a dumbbell-shaped configuration (Figure 8).

Eight patients had lesions localized to the CPA/CMF region (Figure 9). Among them, two had concurrent bilateral lesions. In one bilateral case, the lesions were asymmetric with questionable evidence of spinal cerebrospinal fluid dissemination, whereas another patient exhibited symmetric bilateral lesions.

None of the CPA ATRTs showed evidence of cranial nerve origin at the time of surgical inspection (Figure 9E).

### 3.3. Pineal Region AT/RT

Eleven patients were diagnosed with ATRT in the pineal region, defined in the transverse plane from the posterior third ventricle to the quadrigeminal cistern, and in the vertical plane between the corpus callosum and the superior vermis. Of these, only one patient had a lesion centered in the posterior third ventricle that appeared to arise from the pineal gland (Figure 10A–D).

The remaining ten tumors of the pineal region, which constitute the majority, were found to originate from the superior medullary velum (SMV) [20,21]. Characteristic neuroimaging features included posterior displacement of the superior vermis and fastigium, as well as rostral displacement of the tectum, accompanied by a mass occupying the quadrigeminal cistern and the upper fourth ventricle (Figure 10E,F). Postoperative MRI following the resection of tumors arising from the SMV demonstrated preservation of both the cerebellar vermis and the tectum (Figure 10G,H). A postmortem examination of an ATRT involving the SMV revealed a defect in the roof of the fourth ventricle, with intact superior cerebellar peduncles and tectal plate—findings that strongly suggest an origin from the SMV (Figure 10I). Among the ten patients with SMV ATRT, eight exhibited central necrosis or varying degrees of cyst formation, while one patient presented with evidence of intra-tumoral hemorrhage. All patients with ATRT in the pineal region experienced hydrocephalus, and only one patient showed MRI evidence of subarachnoid dissemination at the time of diagnosis.

The mean age at diagnosis for patients with pineal region AT/RT was 12.4 months, with most cases occurring in infancy (approximately 9 months).

### 3.4. Spine AT/RT

Among the six patients diagnosed with ATRT of the spine, ages ranged from 7 months to 134 months, with a median age of 85 months. Notably, the median age at diagnosis was 80 months, which is older than that of patients with intracranial ATRT.

Only one case involved an intramedullary tumor located in the cervical cord of a 34-month-old girl (Figure 11A–D). Two patients, a 64-month-old girl and a 96-month-old girl, had intradural extramedullary tumors situated at the T12-L1 and L3-S2 levels, respectively. Additionally, a 7-month-old girl presented with a contiguous intradural extramedullary and extradural tumor that extended outward to the paraspinal structures (Figure 11E–I). She underwent a staged operation, initially through a cervical laminoplastic laminotomy at the C5–7 level, followed by a second surgery via an anterior cervical approach for the total resection of the extraspinal disease after chemotherapy.

Another child, the oldest in this cohort, was an 11-year-old boy who presented with a dumbbell-shaped intradural extramedullary ATRT extending into the extradural space and neural foramina at the C5-T11 levels. Additionally, a 9-year-old boy had ATRT located in the epidural space, extending into the neural foramina at the T7–9 levels (Figure 11J–M).

### 3.5. Cases of Rhabdoid Tumor Predisposition Syndrome

Two sibling pairs with ATRT were identified in this cohort. In the first family, the sister harbored tumors in the fourth ventricle and cerebellopontine angle (CPA), while her brother had an intraventricular tumor in the lateral ventricle. Molecular testing in their father with a history of schwannomatosis revealed a germline SMARCB1 variant. In the second family, two sisters were each diagnosed with posterior fossa ATRT, and their deceased maternal uncle had a posterior fossa ATRT as well. This strong familial aggregation, depicted in the pedigree, met criteria for rhabdoid tumor predisposition syndrome (RTPS) [22]. Overall, only four children in this series (8%) were diagnosed with RTPS.

### 3.6. Postoperative Managements

Histopathological examinations conducted by our pathology department confirmed the diagnosis of atypical teratoid rhabdoid tumors (ATRT). However, molecular subgrouping was not available for this cohort in this retrospective review.

All patients in this cohort underwent surgical resection of their tumors. With the exception of two patients—one who died early in the postoperative period due to preexisting coma and another who succumbed to rapid progression of residual disease—all received adjuvant oncologic therapy according to national protocols.

Earlier patients in this cohort were treated according to the Children’s Cancer Group (CCG) study CCG-9921 protocol, which randomized subjects to one of two intensive platinum-based multi-agent chemotherapy regimens followed by maintenance chemotherapy, with the goal of avoiding radiation therapy.

Subsequently, patients were treated on DFCI-AT/RT (NCT00084838), a multi-institutional trial led by Dana-Farber Cancer Institute. This regimen consisted of nearly one year of multi-agent systemic anthracycline-based chemotherapy with concurrent intrathecal therapy. After an initial six-week induction phase, patients received radiation therapy administered concurrently with chemotherapy.

Since 2010, the Children’s Oncology Group (COG) trial ACNS0333, a prospective protocol specifically designed for ATRT, has been used. This regimen includes two cycles of intensive multi-agent chemotherapy incorporating high-dose methotrexate, followed by three tandem cycles of high-dose chemotherapy with autologous stem cell rescue and involved-field radiation therapy. The timing of radiation is determined by patient age as well as disease location and extent.

## 4. Discussion

ATRTs in children exhibit remarkably broad topographic distribution within the CNS, occurring in intraparenchymal, intraventricular, and subdural compartments, and less commonly in epidural locations of both the brain and spine. This breadth of anatomic involvement is unusual among primary pediatric CNS tumors. Consistent with their histologic diversity, ATRTs commonly demonstrate marked imaging heterogeneity, frequently containing cystic components, necrosis, and hemorrhagic foci. Contrast enhancement on MRI is variable and often inhomogeneous, ranging from minimal to patchy or irregular enhancement.

Operative observations in our cohort indicate that intracranial ATRTs predominantly arise from brain parenchyma rather than from the dura or calvarial structures. Site-specific patterns of origin appear plausible based on imaging–surgical correlation: selected pineal region ATRTs likely originate from the superior medullary velum; CPA ATRTs extending from the anterolateral cerebellar hemisphere; and fourth ventricle–CPA junction tumors from the inferior vermis. In the spine, extramedullary ATRTs frequently involve nerve roots—most often dorsal (posterior) roots—and may present in intradural, extradural, or extraspinal compartments. By contrast, intramedullary spinal ATRTs, although rare, arise within the spinal cord parenchyma itself.

The anatomic predilections described above are derived from cohort-level observations and should be considered emerging patterns rather than definitive rules, given the biological heterogeneity of ATRT. Nearly half of ATRT cases present with almost invariably large CNS tumor within the first year of life, supporting a congenital origin in a substantial subset. To better delineate ATRT histogenesis, integrated clinicoanatomic correlations grounded in developmental neurobiology—particularly patterns of embryologic aberration—will be informative.

### 4.1. SMARCB1

Cytogenetic analyses of rhabdoid tumors predominantly reveal abnormalities on chromosome 22, including monosomy, deletions, and translocations, particularly involving 22q11. These genetic alterations indicate the loss or inactivation of a crucial tumor suppressor gene implicated in the initiation and progression of rhabdoid tumors [23].

Versteege et al. (1998) identified SMARCB1 (hSNF5/INI1) as a tumor suppressor by discovering truncating mutations in aggressive pediatric malignancies [24]. They demonstrated a uniform loss of SMARCB1 protein in malignant rhabdoid tumors (MRTs), characterized by the classic pattern of germline loss of one allele followed by the somatic loss of the second allele. The deletion or mutation of SMARCB1 at 22q11.2 is consistently observed in MRTs and is considered a primary oncogenic driver, although the mechanisms initiating these changes remain unclear.

Biegel et al. (1999) evaluated SMARCB1 as a candidate tumor suppressor in atypical teratoid/rhabdoid tumors (ATRT) across a cohort of 29 MRTs, which included 18 ATRTs [25]. They found homozygous deletions of one or more INI1 exons in 15 tumors and diverse mutations in 14. Germline INI1 mutations were detected in four children, further supporting the central role of INI1/SMARCB1 in MRT pathogenesis.

In a microarray gene expression study, Gadd et al. (2010) proposed that rhabdoid tumors may arise from the developmental arrest of neural crest stem cells [26]. They noted significant downregulation of genes involved in neural and neural crest development in rhabdoid tumor samples and concluded that early progenitor cells, during a critical developmental window, are particularly susceptible to SMARCB1 loss. This loss leads to repression of neural differentiation and dysregulation of cell-cycle control.

### 4.2. Role in Chromatin Remodeling

Chromatin remodeling modulates DNA accessibility by repositioning or restructuring nucleosomes, enabling replication, repair, and transcription. The ATP-dependent Switch/Sucrose Non-Fermentable (SWI/SNF) complex—known in mammals as the Brg1/Brm-associated factor (BAF) complex—engages nucleosomal DNA and histone tails to increase DNA accessibility for transcription factors. In humans, the BAF complex interacts and remodels nucleosomes by binding to nucleosomal DNA and histone tails, allowing DNA more accessible to transcription factors, promoting gene expression [27].

Human BAF complexes are assembled from at least 29 genes across 15 families, generating extensive compositional and functional diversity, including tissue-specific subunits [28]. A neural progenitor-specific BAF complex forms in neural stem cells and is distinct from the pluripotent embryonic stem cell BAF (esBAF) complex [29]. Through nucleosome positioning and higher-order chromatin architecture, BAF complex orchestrates gene expression; mutations that perturb these activities can disrupt developmental programs and drive tumorigenesis.

The BAF complex includes SMARC family ATPases—SMARCA4 (BRG1) and SMARCA2 (BRM)—and core subunits such as SMARCB1, SMARCC1, and SMARCC2, alongside cell-type-specific variants. Precise subunit composition is critical for maintaining pluripotency, directing neural differentiation, and guiding tissue development. In embryonic stem cells, epigenetic modifiers and chromatin-remodeling assemblies, including BAF, localize to active promoters and enhancers [30]. Combinatorial subunit assembly underlies context-specific functions of SMARCB1 in tissue development, including the cerebellum [31].

### 4.3. BAF Complex and SMARCB1

SMARCB1 (INI1), expressed in all normal cell nuclei, is essential for chromatin remodeling and gene regulation; its loss disrupts nucleosome remodeling and chromatin control, driving malignancy. Biallelic inactivation of SMARCB1 (or, less commonly, SMARCA4) is a diagnostic hallmark of ATRT. SMARCB1 was the first BAF subunit identified as mutated in cancer and functions as a potent tumor suppressor: Smarcb1-null mice die early in embryogenesis (embryonic day 3.5–5), whereas Smarcb1-heterozygous mice rapidly develop aggressive tumors, including rhabdoid-like neoplasms (median onset ~11 weeks) [32]. The rapid onset and near-complete penetrance of cancer following SMARCB1 inactivation underscore its potent tumor suppressor role.

BAF subunits are mutated in over 20% of human cancers [33,34]. While BAF alterations occur across diverse malignancies and typically in older patients, ATRT in infants and young children is most often driven by biallelic SMARCB1 loss—contrasting with the monoallelic tumor-suppressor patterns common in other cancers [29]. One proposed oncogenic mechanism is the failure of mutant BAF to antagonize Polycomb Repressive Complex 2 (PRC2), which mediates H3K27me3 to repress transcription [35]. Notably, MRT/ATRT genomes exhibit a striking paucity of mutations beyond SMARCB1 loss, among the lowest mutational burdens reported in human tumors [36].

During development, BAF subunits are dynamically regulated, including mutually exclusive paralogs such as ACTL6A (expressed in neuronal progenitors, supporting self-renewal) and ACTL6B (required for neuronal differentiation). Paralog subunits of the BAF complex are central regulators of lineage specification. In the developing cerebellum, granular neuron precursors express ACTL6A and differentiated granular neurons express ACTL6B in a mutually exclusive manner. In malignant rhabdoid tumors (MRTs), aberrant co-expression of ACTL6A and ACTL6B is observed, accompanied by altered DNA methylation at differentiation-associated genes and at the neuronal marker NeuN. Notably, tumors lacking both ACTL6A and ACTL6B tend to show greater epithelial and mesenchymal differentiation [37].

### 4.4. DNA Methylation

DNA methylation profiling enhances CNS tumor diagnostics and molecular classification, defining subgroups with distinct clinical and biological features [38]. During fetal brain development, targeted demethylation activates genes required for neural differentiation, whereas pluripotent stem cells remain globally hypermethylated—mirroring the low-differentiation state observed in ATRT. In ATRT, neural transcriptional regulators preferentially target hypermethylated regions associated with PRC2, leading to repression of neural developmental genes [39]. Although PRC2 typically represses transcription via H3K27me3, ATRTs exhibit depleted H3K27me3 levels, suggesting an H3K27me3-independent epigenetic mechanism in ATRT pathogenesis [40]. Consistent with this, Pekkarinen et al. proposed that pervasive DNA hypermethylation suppresses neural differentiation by silencing key regulators (e.g., NEUROG1, NEUROD2) that cannot bind methylated DNA [40].

### 4.5. Molecular Subgrouping

Torchia et al. (2015) identified two ATRT subgroups based on ASCL1 expression [41]. ASCL1 is a DNA-binding transcription factor that partners with cofactors to regulate gene expression and is essential for neurogenesis, both initiating neuronal differentiation and maintaining progenitor pools. ASCL1-positive (Group 1) tumors are enriched for brain development and NOTCH signaling genes and are associated with better survival, whereas ASCL1-negative (Group 2) tumors show enrichment for mesenchymal differentiation and BMP signaling and portend poorer outcomes. The authors further refined this framework into three epigenetic groups—Group 1 and Group 2 subdivided into 2A and 2B—integrating genomic profiles, SMARCB1 genotypes, and chromatin landscapes [42]. Group 1 tumors are predominantly supratentorial (73.1%; median age 24 months), Group 2A primarily infratentorial (65.6%; median age 12 months), and Group 2B broadly distributed and typically older (>3 years).

Johann et al. (2016) subsequently defined three robust molecular subgroups—ATRT-TYR, ATRT-SHH, and ATRT-MYC—using DNA methylation and gene-expression profiling, aligning closely with Torchia’s schema (ATRT-SHH ≈ Group 1; ATRT-TYR ≈ Group 2A; ATRT-MYC ≈ Group 2B) [7]. Despite shared core genetics, these subgroups exhibit marked differences in DNA methylation, transcriptomes, SMARCB1 mutation spectra, clinical features, anatomic predilections, and neuroimaging characteristics. The three-subgroup classification (ATRT-TYR, ATRT-SHH, ATRT-MYC) is now widely adopted [13,43].

ATRT-SHH comprises roughly half of ATRTs and is heterogeneous by age and location. Federico et al. subdivided ATRT-SHH into SHH-1A, SHH-1B, and SHH-2 [44], which differ in age at onset and anatomic distribution: SHH-1A (median 18 months; 88% supratentorial), SHH-1B (median 107 months; 85% supratentorial), and SHH-2 (median 13 months; 93% infratentorial, often extending to the pineal region). Germline SMARCB1 variants are enriched in SHH-2; ASCL1 expression is enriched in SHH-1B; OLIG2 and GFAP are typically absent in SHH-2. Older SHH-1B patients tend to have more favorable outcomes.

Overall, ATRT-SHH accounts for ~52% of ATRTs (SHH-1A 25%, SHH-1B 17%, SHH-2 10%), with ATRT-TYR at ~27% and ATRT-MYC at ~17% Most ATRTs show loss of SMARCB1/INI1 expression; a smaller subset shows loss of SMARCA4/BRG1 (ATRT-SMARCA4, ~0.5–4%) [45,46,47].

ATRT-SMARCA4 retains INI1 expression, exhibits distinct methylation and RNA profiles, predominantly affects infants (median 3 months), carries a high prevalence of germline mutations, and tends to be clinically more aggressive [46]. Loss of nuclear expression of INI1 protein is typically due to deletion or mutation of the SMARCB1 locus on chromosome 22q11.2, whereas alterations at the SMARCA4 locus are located on chromosome 19p13.2. ATRT-SMARCA4 is extremely rare, and no cases were found in this cohort.

### 4.6. Metastases of ATRT

Metastases of ATRT, either positive for CSF cytology or disclosed on neuroimaging at diagnosis, are fairly common. According to St. Judes experience, 18 out of 52 (34.6%) infant ATRT (<1 year old), had M+ disease while 8 of 22 (36.4%) had M+ disease before treatment, the rate of metastases are higher among ATRT-SHH subgroups [19]. CSF dissemination reflects aggressive biology and complicates management, influencing both surgical strategy and radiotherapy planning: for localized disease, clinicians may pursue maximal safe resection with involved-field radiation, whereas disseminated disease often necessitates more conservative surgery and craniospinal irradiation.

### 4.7. Rhabdoid Tumor Predisposition Syndrome (RTPS)

Rhabdoid tumor predisposition syndrome (RTPS) is a rare autosomal-dominant condition that predisposes to malignant rhabdoid tumors (MRTs), including renal rhabdoid tumors (RTK), atypical teratoid/rhabdoid tumors (ATRT), and extracranial extrarenal rhabdoid tumors (EERT) [46].

Two genetic forms are recognized: RTPS1, caused by germline SMARCB1 variants, and RTPS2, caused by germline SMARCA4 variants. Diagnosis integrates clinical evaluation (including detailed family history), immunohistochemistry demonstrating loss of SMARCB1 or SMARCA4 protein expression, and tumor sequencing for somatic alterations; germline testing for pathogenic variants in SMARCB1 or SMARCA4 is recommended for all patients with ATRT. Synchronous or metachronous tumors in RTPS often show heterogeneous DNA-methylation profiles, supporting non-clonal origins [10]. RTK is the most frequent synchronous tumor in RTPS [48].

One family in this cohort with RTPS1 had a notable history of paternal schwannomatosis and infantile ATRT in his children. Germline mutations in the *SMARCB1* gene are implicated in both infantile ATRT and adult schwannomatosis, although the two conditions differ in mutation type, tumor phenotype, clinical behavior, and penetrance. In ATRT, mutations typically involve truncating or null/near-null alleles, leading to an aggressive, high-grade malignancy with high penetrance. By contrast, mutations associated with adult schwannomatosis are usually hypomorphic or otherwise less disruptive, resulting predominantly in benign schwannomas with lower penetrance and a later age of onset.

Germline SMARCB1 mutations (RTPS1) are present in 25–35% of ATRT patients, who tend to be younger with more extensive disease [43,49,50]; the penetrance of RTPS1 is high, approximately 90% by age 5 [51]. Median age at diagnosis for RTPS-associated MRTs is 4–7 months versus 13–30 months for sporadic cases.

The genetic diagnosis of RTPS involves clinical assessment, particularly focusing on family history, along with immunohistochemical results indicating SMARCB1/SMARCA4 deficiency, in addition to screening for somatic mutations [22]. Molecular genetic testing to identify a germline heterozygous pathogenic variant in SMARCB1 or SMARCA4 is recommended for all cases of ATRT diagnosis. Notably, around 66% of congenital cases (<28 days) harbor germline SMARCB1/SMARCA4 variants [12]. The children with RTPS have poorer survival with 5-year overall and event-free survival rates are ~20% and ~15%, respectively; adverse prognostic factors include age <1 year, EERTs, metastatic presentation, and presence of synchronous tumors (seen in one-third of RTPS) [12,13,14]. Adults with germline variants may develop multiple schwannomas.

### 4.8. Histogenesis

Although genetically and histologically well-characterized, malignant rhabdoid tumors (MRTs)—including atypical teratoid rhabdoid tumors (ATRTs)—exhibit significant intratumoral heterogeneity, and the precise cells of origin remain a subject of debate [52,53,54]. ATRTs are thought to originate from embryonal cells that exhibit stalled differentiation, similar to other pediatric embryonal tumors. Additionally, the molecular similarities between CNS ATRTs and extracranial MRTs suggest that the potential cells of origin may extend beyond CNS-restricted precursors.

Torchia et al. described two molecular classes within Groups 1 and 2, distinguished by ASCL1 expression and enriched for specific neurogenic and forebrain markers (LHX, MEIS, FABP, ASCL1), hindbrain markers (ZIC2, OTX2), or mesenchymal markers (BMP4, MSX1) [41]. Group 1 tumors show enrichment in the proneural NOTCH signaling pathway and genes that regulate neural differentiation, while Group 2 tumors are characterized by enrichment in BMP signaling and pathways related to cell adhesion and migration. The distinct transcriptional signatures and anatomical preferences of these molecular subtypes suggest differing cellular origins, with lineage-specific loss of SMARCB1 and additional alterations involving unknown modifier genes contributing to the clinical heterogeneity observed in these tumors [41].

In search for histogenesis of ATRTs, it is reasonable to propose that specific mutations occur in particular progenitor cells. Neural stem cells (NSCs) and neural crest cells (NCCs) have been considered as potential candidates for the histogenesis of ATRTs.

### 4.9. Neural Stem and Neural Progenitor Cells

Embryonic stem cells, which are pluripotent, can differentiate into a wide range of cell types, including neural stem cells (NSCs) and neural crest cells (NCCs). NSCs have the ability to self-renew indefinitely and generate neurons, astrocytes, and oligodendrocytes. In contrast, neural progenitor cells (NPCs)—which are the daughter cells derived from NSCs—exhibit more limited self-renewal and proliferative capacity [55]. SMARCB1 plays a critical role during neural differentiation, particularly in the transition from NSCs to NPCs. The loss of SMARCB1 is lethal prior to the NSC stage but has less severe consequences at more mature stages [55]. Neurulation, occurring between gestational weeks 3 and 4 in human, forms the neural tube from the neural plate, thereby initiating CNS development.

### 4.10. Neural Crest Cells

Neural crest cells (NCCs) arise at the neural plate border during primary neurulation. These cells represent a multipotent, transient embryonic cell population located at the lateral border of the neural plate, situated between the neuroectoderm and non-neural ectoderm. As the neural plate folds inward to form the neural tube during primary neurulation, NCCs are initially positioned in the neural folds near the fusion point.

During neurulation, boundary cells come into proximity, with the edge cells of the non-neural ectoderm facilitating the zipping and fusion at the dorsal midline of the neural tube [56]. After the closure of the neural tube by the end of the fourth week of gestation, NCCs undergo epithelial–mesenchymal transition (EMT), detaching from the neural tube and transforming into migratory mesenchymal cells.

NCCs migrate extensively throughout the developing embryo along specific pathways, guided by signaling molecules and interactions with surrounding tissues. They comprise distinct regions, including cranial, vagal, trunk, and sacral components, each exhibiting unique migratory behaviors and fates. NCCs give rise to diverse cell lineages, including sensory and autonomic neurons, Schwann cells, melanocytes, craniofacial cartilage and bone, smooth muscle, dentin, adrenal medulla, and portions of the meninges.

Importantly, NCCs do not contribute to the CNS parenchyma; instead, neuroepithelial cells—neural stem cells of the neural tube—primarily contribute to neurogenesis and the maintenance of progenitor cells (radial glial cells) in the CNS. The entire CNS arises solely from the neural tube, highlighting the distinct developmental pathways of these cell populations.

NCCs have been proposed as candidate cells of origin for ATRT based on histological heterogeneity and developmental plasticity. Wang and Furnari used human induced pluripotent stem cell (hiPSC)-derived NCCs and found that SMARCB1 knockdown during directed neural differentiation resulted in the production of NPCs with transcriptomic features of ATRT—particularly the SHH subgroup—whereas intact SMARCB1 allowed for normal differentiation [57].

In a separate study on gene expression in RTK signaling, Gadd et al. reported downregulation of neural and neural crest developmental genes in MRTs, which is consistent with early progenitor arrest and SMARCB1-mediated repression of neural development [26]. Custers et al. integrated phylogenetics and single-cell mRNA sequencing in patient-derived organoids and identified shared mutations between tumor and adjacent normal tissues, suggesting lineage derivation from Schwann cells (a neural crest lineage) [54]. Furthermore, the loss of SMARCB1 impedes NCC differentiation along mesenchymal trajectories, as extrarenal MRTs exhibit ectodermal (neural crest) rather than mesodermal lineage signatures [54].

Han et al. generated Smarcb1(flox/flox); Rosa26-Cre(ERT2) mice, which allow for the temporal inactivation of SMARCB1 from embryogenesis through adulthood [52]. Inactivation at embryonic days 6 to 7 resulted in the development of intracranial tumors (in the hippocampus, cerebellum, and cortex) or spinal tumors, which manifested neurological symptoms approximately 90 days later, effectively recapitulating human atypical teratoid rhabdoid tumors (ATRT). Conversely, inactivation at embryonic day 12 did not lead to tumor formation, suggesting the existence of a developmentally restricted susceptible population. Cross-species analyses revealed that murine intracranial ATRTs correlate with neuronal progenitors, while extracranial tumors are associated with ectomesenchyme, a tissue derived from the cephalic neural crest. Murine intracranial tumors correlate more with neural progenitors, which aligns with some of human intracranial tumors correlating with neural stem cell markers and genes such as NOTCH, FABP7, POU3F2, SOX2, and MYCN. Collectively, these data support at least two origins: intracranial tumors from neural progenitors and extracranial tumors from neural crest-derived lineages, consistent with human heterogeneity and subgroup-specific signatures [52]. The observed heterogeneity in human ATRTs, characterized by variable subgrouping, likely arises from different stem or progenitor cells.

### 4.11. ATRT Subgroups and Candidate Cells/Anatomy

ATRTs exhibit significant anatomical, histological, and molecular heterogeneity. Their distribution correlates with distinct molecular subgroups—TYR, SHH (subclassified into SHH-1A, SHH-1B, and SHH-2), and MYC—which collectively inform hypotheses about their developmental origins and clinical behavior.

#### 4.11.1. ATRT-TYR

ATRT-TYR tumors are characterized by the enrichment of melanosomal marker genes such as TYR, DCT and OTX, which are associated with melanogenesis and neural crest development [58,59]. Although the specific role of TYR in ATRT tumorigenesis has yet to be fully elucidated, it is noteworthy that several other components of the melanosomal pathway, including TYRP and MITF, are also overexpressed in ATRT-TYR. This overexpression may reflect the restricted neuroectodermal origins of these tumors [13]. The expression patterns of these melanosomal genes suggest that the cell of origin could be a NCC committed to the melanosomal lineage. However, due to the lack of in vivo and in vitro models of ATRT-TYR, the mechanisms underlying tumor development remain unclear [55].

Most ATRT-TYR tumors are infratentorial with only 18–25% being supratentorial [13,58]. Lobón-Iglesias et al. proposed their origins in the middle cerebellar peduncle and inferior cerebellar vermis [59].

#### 4.11.2. ATRT-SHH

ATRT-SHH tumors exhibit overexpression of axonal guidance genes (SEMA6A, TUBB2B/3/4A) and neural developmental genes (FABP7, LHX2, MEIS2). Additionally, genes involved in the SHH pathway (MYCN, GLI2, PTCH1, BOC) and components of the NOTCH pathway (ASCL1, HES5/6, DLL1/3, DTX1) are also overexpressed, supporting their classification as neural in origin [55]. These tumors likely arise from NPCs.

ATRT-SHH tumor arises in both supratentorial and infratentorial compartments, with approximately 30% localizing to the pineal region and often extending across both compartments [47,58,60]. Drawing on cross-species correlations, Lobón-Iglesias et al. linked human imaging phenotypes to developmental niches in mouse models [59]: the cerebellar anterior lobe at the midbrain–hindbrain boundary likely the origin of infratentorial SHH ATRT, while the ganglionic eminence (GE) appears to give rise to basal ganglia and intraventricular SHH ATRT in the supratentorial location. The GE is a transient, progenitor-rich structure in the ventral telencephalon that drives basal ganglia and cortical development via tangential migration; molecular signatures in basal ganglia/intraventricular SHH ATRT support this origin [59].

Within ATRT-SHH, subgroups exhibit distinct anatomic predilections and transcriptional programs [45,59], with SHH-1 predominating in the supratentorial compartment and SHH-2 in the infratentorial compartment [61]. In keeping with these patterns, SHH-2 aligns with cerebellar anterior-lobe tumors characterized by EN2 overexpression, whereas SHH-1A aligns with basal ganglia/intraventricular tumors marked by OLIG2 overexpression [45,59].

#### 4.11.3. ATRT-MYC

ATRT-MYC epigenetically clusters with extracranial MRTs, characterized by global DNA hypomethylation and overexpression of HOX and mesenchymal programs, pointing to a shared—or non-CNS—cell of origin [59,62,63]. Single-cell RNA comparisons with the developing mouse brain further support this non-CNS hypothesis [61,64]. Schwann cell precursors have been implicated as candidate cells of origin for ATRT-MYC; however, canonical Schwann markers (e.g., SOX10) are often absent in cranial nerve ATRTs, suggesting lineage derivation from neural crest-related precursors with divergent differentiation rather than mature Schwann cells per se [24,26,39,65].

Consistent with this, a subset of ATRT-MYC arises from extra-parenchymal sites, particularly along cranial nerves, mirroring human presentations [66,67,68,69]. Mouse models of extra-parenchymal ATRT originating from cranial nerves and peripheral brain regions implicate neural crest cell precursors as the lineage of origin [59]. Collectively, these data indicate that ATRT-MYC tends to occur in non-parenchymal CNS structures and is best explained by neural crest-derived lineages [55,59,66,67,68,69].

Graf et al. proposed primordial germ cells (PGCs) as a possible cell of origin for a subset of murine ATRT-MYC tumors [70]. In humans, PGCs arise from epiblast-derived precursors during the second week of embryogenesis and migrate along developing nerve fibers in close association with Schwann cells [71]. While aberrant PGCs are well established in the pathogenesis of germ cell tumors, including teratomas [72], their involvement in ATRT remains speculative and controversial [55,59].

### 4.12. Topographic Distribution of ATRT

In our cohort, tumor locations were distributed as follows: 36% infratentorial, 31% supratentorial, 24% pineal, and 9% spinal. Reported topographic distributions vary across series. Ho et al. described 60% infratentorial, 32% supratentorial, 4% pineal, and 2% spinal, noting the cerebellum as the most common infratentorial site (63%) [13]. The Children’s Oncology Group reported 51% infratentorial, 40% supratentorial, 7.5% pineal, and 1.5% spinal [14]. Rorke et al. found 56% cerebellar, 26.7% cerebral, 9.3% pineal, 3% suprasellar, and 3% extramedullary cord involvement [5]. Silva et al. reported 66.7% posterior fossa, 17.8% supratentorial, and 11.1% pineal; among posterior fossa cases, the fourth ventricle was the predominant epicenter (63%), with 10% extending into the cerebellopontine angle (CPA) and 27% primarily centered in the CPA [15].

### 4.13. Subgroup–Location Correlations

Precise anatomic location and origin are often difficult to assign preoperatively due to tumor size and extension. They should be assessed based on not only preop imaging but also intraoperative observation and imaging after tumor resection.

Intracranial ATRTs exhibit distinct subgroup–anatomical location patterns. ATRT-TYR predominantly arise infratentorially (~70–75%), whereas ATRT-SHH are mainly supratentorial (~65%), and approximately 50–70% of ATRT-MYC are also supratentorial. Notably, all cranial nerve and spinal ATRTs belong to the MYC subgroup [73], and all analyzed extracranial MRTs likewise classify as MYC [74].

#### 4.13.1. Supratentorial ATRTs

Supratentorial ATRTs arise within the cerebral hemispheres, basal ganglia/thalamus, and ventricular system, though the precise site of origin may be difficult to determine in large lesions. Lobón-Iglesias et al. reported a subtype–location association: basal ganglia and ventricular tumors are predominantly ATRT-SHH, whereas cortical tumors are largely ATRT-MYC [66].

#### 4.13.2. Pineal and Para-Pineal ATRTs

Pineal ATRTs are rare, with only four cases reported in the English literature [75]. Several tumors labeled as “pineal” likely arise from para-pineal structures, particularly the superior medullary velum (SMV) [21]. Developmentally, the SMV forms from the rhombencephalic boundary toward the end of cranial neurulation and constitutes the roof of the fourth ventricle; ATRTs in this location are therefore hypothesized to derive from neural progenitor cells within the SMV. Supporting this regional origin, Federico reported that 17 (63%) of 27 infratentorial ATRT-SHH tumors extended into the midbrain and pineal region, all classified as SHH-2 [44]. Similarly, Lobón-Iglesias et al. noted that ATRT-SHH may arise near the midbrain–hindbrain boundary—potentially within the anterior cerebellar lobe—and can present at or near the pineal region [59].

#### 4.13.3. Infratentorial ATRTs:

The cerebellar peduncular, vermis, and CPA/CMF are common site of ATRTs. Most ATRT-TYR tumors arise from the middle cerebellar peduncle and inferior vermis [76] and can extend exophytically into the cerebellopontine angle (CPA) and cerebellomedullary fissure (CMF). ATRT-MYC tumors can originate from cranial nerves in the CPA, producing imaging appearances that overlap with TYR tumors; however, the age distribution differs, with TYR typically affecting much younger children than MYC. Infratentorial ATRTs frequently mimic posterior fossa ependymomas in their location and propensity to extend into the CPA and fourth ventricle, but ATRTs tend to be more infiltrative, whereas ependymomas are usually expansile and less invasive [77]. Consistent with this pattern, the European Rhabdoid Registry reports cranial nerve involvement occurs in approximately 3% of ATRT cases and is most often associated with the MYC subgroup [62]. One female patient with CPA ATRT of CPA, who was notably the oldest in this cohort, had a tumor that likely originated from the vestibular nerve, without involvement of the brainstem or cerebellum. (Figure 12).

#### 4.13.4. Spinal ATRTs

Spinal ATRTs are uncommon, comprising 2–4% of cases in the literature [6,16], though they accounted for 10% in our cohort. The first primary spinal ATRT was described by Robson et al. in 1987 [78]. Notably, only 1 of 65 CNS ATRTs was spinal in the Children’s Oncology Group ACNS0333 trial. Reported spinal compartments include intradural extramedullary (52.6%), intramedullary (35%), and extradural (24.6%), with approximately 20% spanning both intra- and extradural compartments [79]. Involvement by vertebral level includes lumbar (60%), thoracic (55%), cervical (32.8%), and sacral (20%), and more than half of cases affect multiple levels [78,79]. Patients with spinal ATRTs tend to be older than those with intracranial disease (mean age ~8.5 years in the present cohort) [80]. Most spinal cases belong to the ATRT-MYC subgroup and frequently involve spinal nerve roots [7,13,61], consistent with peripheral nerve–lineage features.

Table 1 summarizes concept of topographic distribution and histogenesis of each subgroup of ATRT.

## 5. Conclusions

Atypical teratoid/rhabdoid tumors (ATRTs) predominantly affect very young children; in our cohort, 44% were diagnosed within the first year of life. They arise throughout the CNS—including supratentorial and infratentorial compartments, the pineal region, and the spinal cord—and may be intra- or extra-parenchymal, occasionally with involvement of extradural spaces. This broad anatomic distribution and variable compartmental involvement distinguish ATRT from other pediatric CNS tumors.

### 5.1. Molecular Subgroups and Clinical Correlates

Comprehensive genomic, transcriptomic, and DNA methylation profiling has established three principal ATRT subgroups—TYR, SHH, and MYC—that have gained broad acceptance in recent years.

ATRT-TYR predominantly occurs in infratentorial locations, likely arising from the cerebellar peduncle and inferior vermis, and predominantly affects very young patients. In contrast, the SHH subgroup is further divided by specific methylation profiles into SHH-1A, SHH-1B, and SHH-2. SHH-1A tends to occur at supratentorial sites with a younger average age, whereas SHH-1B also presents supratentorially but in significantly older children among subgroups. It is hypothesized that supratentorial ATRT may arise from neural stem cells within the ganglionic eminence during neurulation, particularly affecting deep cerebral structures such as the basal ganglia and ventricles. Conversely, the SHH-2 is predominantly found in an infratentorial location in young children and often extends toward the pineal region, where the superior medullary velum is a likely source.

The MYC subgroup encompasses all extracranial MRT, including those arising in peripheral nerves and the spine, and skews toward older patients, with a median diagnostic age around 27 months. Intracranially, MYC tumors often involve cranial nerves but can also occur in the cerebral cortex.

The CPA/CMF and fourth ventricle represent frequent sites for ATRT. Distinguishing the tumor’s origin—whether it arises from the cerebellar peduncle, cerebellum, or cranial nerves with exophytic extension—can be complex; the former typically indicates an ATRT-TYR tumor, whereas the latter suggests an ATRT-MYC tumor.

### 5.2. Genetics and Developmental Timing

Loss or mutation of SMARCB1 is central to ATRT tumorigenesis, although the precise cause and timing of SMARCB1 mutation remain unknown. Germline alterations occur in 25–35% of patients and are associated with earlier onset and more advanced disease. Whether ATRT cytogenesis originates during neurulation from neuroepithelial or neural progenitor populations or after neural crest delamination remains unresolved. Neural stem cells predominantly aligned toward CNS lineages (neurons, astrocytes, oligodendrocytes), whereas neural crest cells contribute mainly to peripheral and mesenchymal lineages, without participating in the formation of brain parenchyma, except for non-parenchymal cranial structures that interact with the brain.

### 5.3. Hypotheses on the Cell(s) of Origin

The pronounced histologic and molecular heterogeneity observed in ATRTs supports competing or complementary hypotheses for their origin. One possibility is an origin in neural crest-derived progenitors, which could account for diverse tissue elements and the peripheral nerve and spinal presentations typical of the MYC subgroup. An alternative or complementary view posits that parenchymal infant brain ATRTs reflect transformation of a distinct subset of neural plate boundary progenitors with bi-potential features contributing to both CNS and neural crest lineages during early embryogenesis.

Given the presence of overlapping lineage markers and subgroup-specific patterns, it is unlikely that a single cell of origin can be identified for all atypical teratoid rhabdoid tumors (ATRTs). To definitively elucidate these origins, an integrated approach involving lineage tracing, temporal developmental modeling, and comprehensive molecular profiling across human tissues and model systems is essential.

The integrated atlases of the human fetal brain have greatly advanced our understanding of fetal brain development at the cellular level, particularly through single-cell genomic approaches such as single-cell RNA sequencing (scRNA-seq). By comparing ATRT tumor cells with single-cell atlases of the developing fetal brain, researchers have begun to identify potential cells of origin for ATRT [64,81]. Furthermore, cross-species scRNA-seq analyses that integrate human tumor data with knockout mouse models have been used to investigate the putative cells of origin for distinct ATRT subtypes [59]. Together, these single-cell genomic studies, which draw on both human fetal brain datasets and ATRT samples, are essential for deepening our understanding of ATRT biology and its developmental origins. Until such comprehensive data are available, any conclusions regarding the cytogenesis of ATRT should be regarded as provisional.

## Figures and Tables

**Figure 1 cancers-18-00008-f001:**
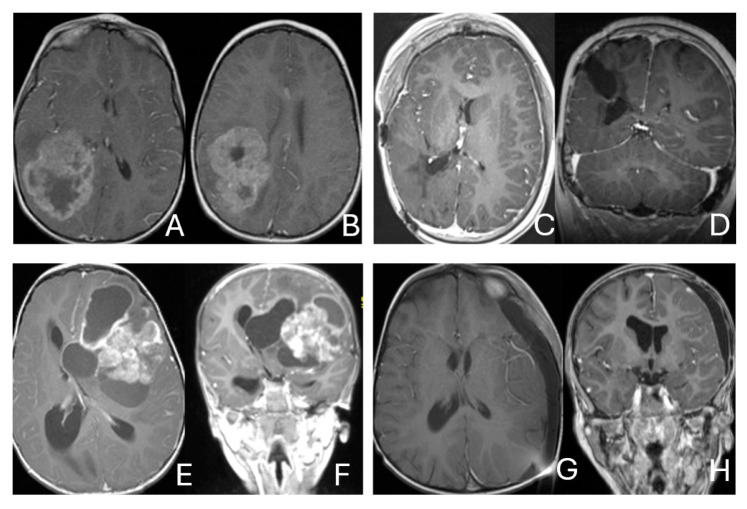
Cerebral lobar ATRT. (**A**–**D**): 20-month-old boy with a parietal intraparenchymal ATRT. Preoperative axial images (**A**,**B**) and postoperative axial (**C**) and coronal (**D**) images confirm the intraparenchymal location following resection. (**E**–**H**): 26-month-old boy with a large heterogeneous frontal mass with multiple peritumoral cysts. Preoperative axial (**E**) and coronal (**F**) images, and postoperative axial (**G**) and coronal (**H**) images after gross-total resection support a frontal lobe origin despite of the basal ganglia origine as the preoperative imaging suggetsed.

**Figure 2 cancers-18-00008-f002:**
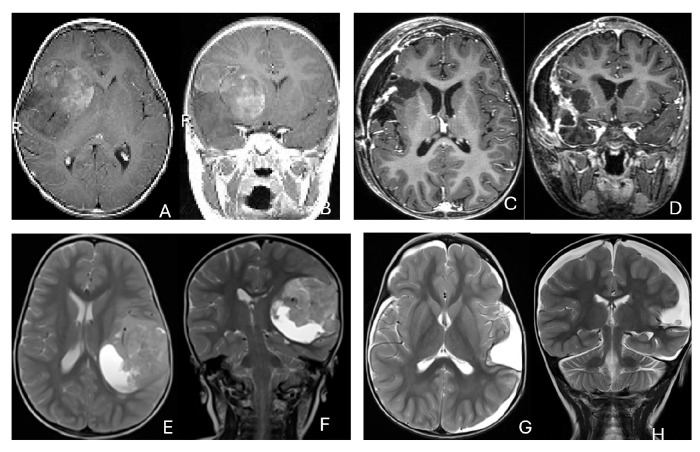
Multi-lobe ATRT. (**A**–**D**): A 3-year-old boy with a heterogeneous frontotemporal ATRT with encasement of Sylvian vessels on MR images ((**A**), axial; (**B**), coronal). Intraoperatively, the tumor replaced both the frontal and temporal opercula, extended to the anterior, and displaced/encased branches of the middle cerebral artery. Postoperative MR images after resection ((**C**), axial; (**D**), coronal) confirm multilobe involvements. (**E**–**H**): of a 30-month-old boy with a temporoparietal cystic ATRT crossing the Sylvian fissure shown on T2-weighted MR images ((**E**), axial; (**F**), coronal). Correlative intraoperative findings and imaging demonstrated the tumor extending from the temporal lobe into the posterior frontal lobe across the Sylvian fissure and encasing middle cerebral artery branch. Post-resection MR images ((**G**), axial; (**H**), coronal) show a resection cavity spanning the temporal and posterior frontal opercula.

**Figure 3 cancers-18-00008-f003:**
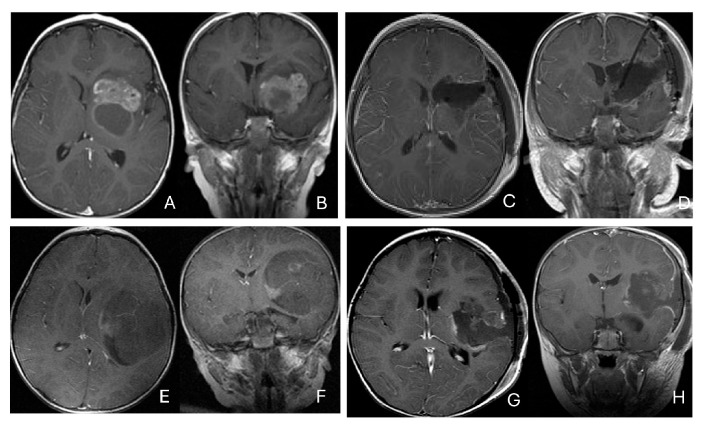
Basal ganglia ATRT. (**A**–**D**): Post contrast MR images ((**A**)-axial, (**B**)-coronal) of 13-month-old boy with deep seated ATRT. Note the solid tumor with peritumoral cysts at the basal ganglia with one cyst extending to the thalamus. Post-resection MR images ((**C**)-axial, ((**D**)-coronal) show the tumor resected cavity in the basal ganglia with the thalamus clear of tumor involvement. (**E**–**H**): Post contrast MR images ((**E**)-axial, (**F**)-coronal) of a 23-month-old boy with a large ATRT of deep temporal lobe and the insula extending to the basal ganglia. The tumor was predominantly in the basal ganglia as shown postoperative MR ((**G**)-axial, (**H**)-coronal).

**Figure 4 cancers-18-00008-f004:**
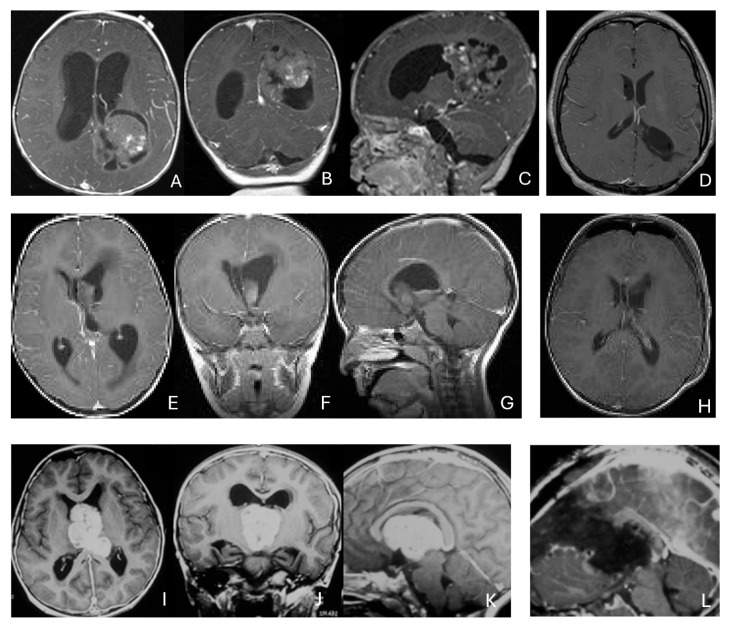
Intraventricular ATRT. (**A**–**D**): A 7-month-old boy with ATRT within the left lateral ventricle with associated hydrocephalus shown on MR images ((**A**), axial; (**B**), coronal; (**C**), sagittal). Note irregular peritumoral cysts and ependymal invasion which was confirmed at surgery. Postoperative MR image ((**D**), axial) shows a gross-total resection. The patient had a history of rhabdoid tumor predisposition syndrome. (**E**–**H**): A 34-month-old boy with MR images ((**E**), axial; (**F**), coronal; (**G**), sagittal) showing an intraventricular mass at the anterior horn based on the septum pellucidum. ATRT was originated at the junction of the septum pellucidum. Post-contrast MR ((**H**), axial) confirms resection. (**I**–**L**): Post-contrast MR images ((**I**), axial; (**J**), coronal; (**K**), sagittal) of a 3.5-year-old boy demonstrate a large, enhancing third ventricular ATRT with partial extension into the lateral ventricle. The lesion was removed via an interhemispheric transcallosal approach ((**L**), sagittal).

**Figure 5 cancers-18-00008-f005:**
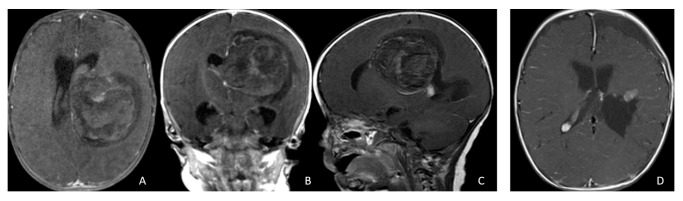
Paraventricular ATRT with ventricular extension. (**A**–**D**): A 3-month-old girl who presented with emesis followed by acute decerebrate posturing. Post-contrast MR images ((**A**), axial; (**B**), coronal; (**C**), sagittal) reveal a large hemorrhagic mass in a deep centrencephalic location. Postoperative MR ((**D**), axial) shows a focal dilation of the lateral ventricle after subtotal resection of a parietal para-ventricular ATRT.

**Figure 6 cancers-18-00008-f006:**
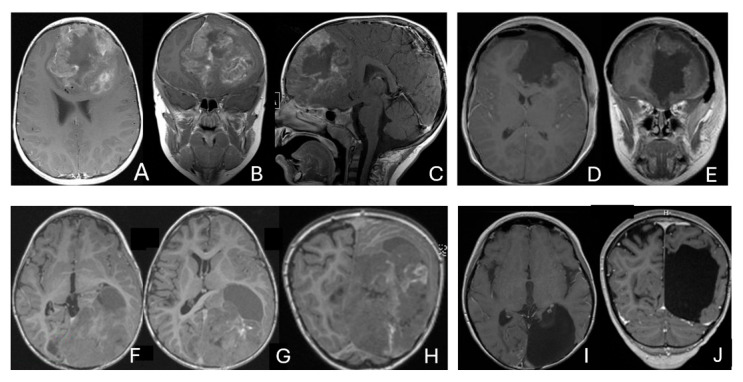
Bi-hemispheric ATRT. (**A**–**E**): MR images ((**A**), axial; (**B**), coronal; (**C**), sagittal) of a 4-year-old boy show a bifrontal ATRT crossing the falx cerebri, with heterogeneous enhancement. The corpus callosum is compressed posteriorly. At surgery, left frontal component invades and permeates through the falx with extension into the right subdural space. Postoperative images after bifrontal craniotomy ((**D**), axial; (**E**), coronal) show tumor resection. Bi-occipital hemispheric ATRT. (**F**–**J**): MR images ((**F**,**G**), axial; (**H**), coronal) of a reveal a 22-month-old girl large, inhomogeneous left occipital lobe mass crossing the midline into the right occipital lobe. The lesion invades the posterior falx cerebri and extends into the medial occipital lobe of the right side. A highly vascular tumor was resected via a left occipital craniotomy; the right occipital component was not removed with concern of blindness due to preexisting right homonymous hemianopia. She received chemoradiation with durable disease resolution for over 15 years, as shown on follow-up post-contrast MR ((**I**), axial; (**J**), coronal).

**Figure 7 cancers-18-00008-f007:**
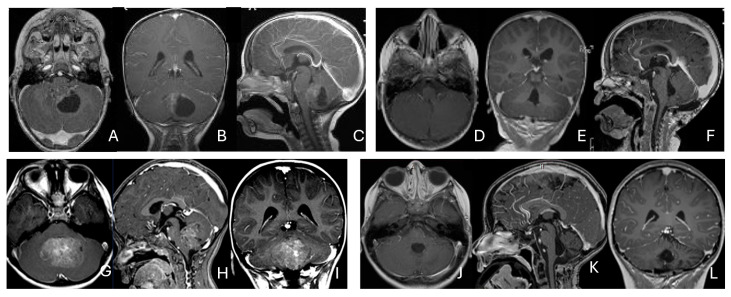
Fourth ventricle ATRT. (**A**–**F**): MR images ((**A**), axial; (**B**), coronal; (**C**), sagittal) of a 21-month-old boy show a predominantly cystic fourth ventricular mass. The lesion appeared to arise from the inferior vermis, and the floor of the fourth ventricle was partially infiltrated. Postoperative images ((**D**), axial; (**E**), coronal; (**F**), sagittal) demonstrate gross-total resection. (**G**–**L**): MR images ((**G**), axial; (**H**), sagittal; (**I**), coronal) of a 3.5-year-old girl reveal a solid fourth ventricular tumor with heterogeneous enhancement. The floor of the fourth ventricle was intact, and the lesion appeared to originate from the inferior vermis. Postoperative images ((**J**), axial; (**K**), sagittal; (**L**), coronal) confirm gross-total resection.

**Figure 8 cancers-18-00008-f008:**
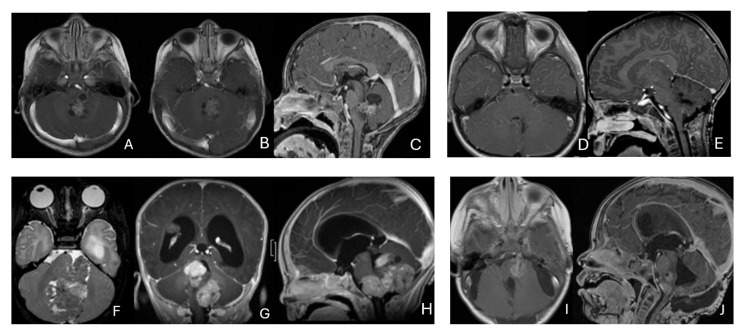
Fourth ventricle-lateral recess/CMFOK ATRT. (**A**–**E**): MR images ((**A**), axial; (**B**), coronal; (**C**), sagittal) of a 12-month-old boy show a fourth ventricular ATRT extending to the CMF through the lateral recess and with multiple peritumoral cysts. A partial invasion to the pons was noted at surgery. Postoperative images ((**D**), axial; (**E**), sagittal) demonstrate gross-total resection. Fourth ventricle-CPA ATRT. (**F**–**J**): MRI ((**F**): axial T2-weighted; (**G**): post-contrast coronal; (**H**): post-contrast sagittal) of a 5-month-old girl reveal a heterogeneous, dumbbell-shaped ATRT extending from the fourth ventricle into the CPA/CMF. During surgery, invasion of the lateral wall of the medulla oblongata was observed, and a subtotal resection was achieved. Post-contrast MRI ((**I**): axial; (**J**): sagittal) reveals residual enhancing tumor at the CMF.

**Figure 9 cancers-18-00008-f009:**
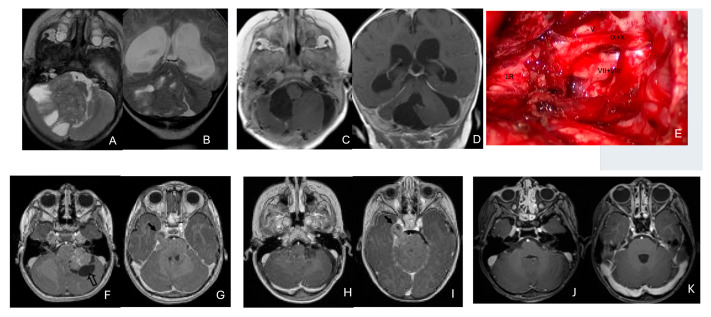
Cerebellopontine angle (CPA) ATRT. (**A**–**E**): T2-weighted MR images ((**A**), axial; (**B**), coronal) of a 7-week-old girl show a massive ATRT centered in the right CPA/CMF. Postoperative MR images ((**C**), axial; (**D**), coronal) show a gross-total resection of a highly vascular, necrotic mass. Intraoperative photograph following tumor resection illustrates key landmarks: LR, lateral recess; V, trigeminal nerve; VII/VIII, facial and vestibulocochlear nerves; IX/X, glossopharyngeal and vagus nerves. The cranial nerves were free of tumor. ATRT appeared to originate from the lateral posterior cerebellar lobe. (**F**–**K**): MR images ((**F**), axial; (**G**), axial) of a 3.5-year-old girl show bilateral CPA tumors: a larger, partially cystic mass on the left (open arrow) and a smaller solid mass extending toward Meckel’s cave on the right (solid arrow). The left-sided tumor was resected via a retrosigmoid approach ((**H**), axial; (**I**), axial). The lesion arose from the lateral cerebellar hemisphere and extended into the CPA cistern without cranial nerve invasion. Following chemoradiotherapy, the right-sided lesion resolved ((**J**), axial; (**K**), axial).

**Figure 10 cancers-18-00008-f010:**
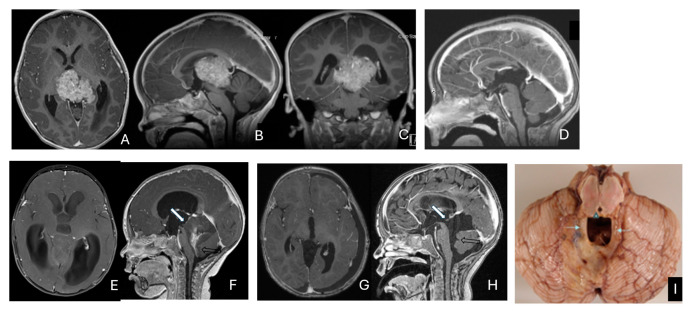
Pineal region ATRT. (**A**–**D**): MR ((**A**), axial; (**B**), sagittal; (**C**), coronal) of a 3-year-old girl show an avidly enhancing ATRT in the posterior third ventricle. Postoperative post-contrast MR ((**D**), sagittal) demonstrates gross-total resection. (**E**–**I**): MR images ((**E**), axial; (**F**), sagittal) of a 13-month-old boy reveals a pineal region tumor extending from the quadrigeminal cistern into the fourth ventricle with obstructive hydrocephalus. The superior vermis and fastigium (open arrow) are depressed, and the tectal plate (solid arrow) is flattened and displaced rostrally. Gross-total resection of largely necrotic ATRT was achieved via a posterior interhemispheric transtentorial approach. Postoperative MR ((**G**), axial; (**H**), sagittal) shows restoration of the tectal plate (solid arrow) and fastigium (arrowhead). The superior vermis and the ependymal lining of the fourth ventricle were not invaded. A representative brain section (**I**) at post mortem study from another patient with a similar presentation demonstrates intact bilateral cerebellar peduncles (arrows) and a preserved tectal plate (arrowhead). Thus, the tumor is considered to arise from the superior medullary velum.

**Figure 11 cancers-18-00008-f011:**
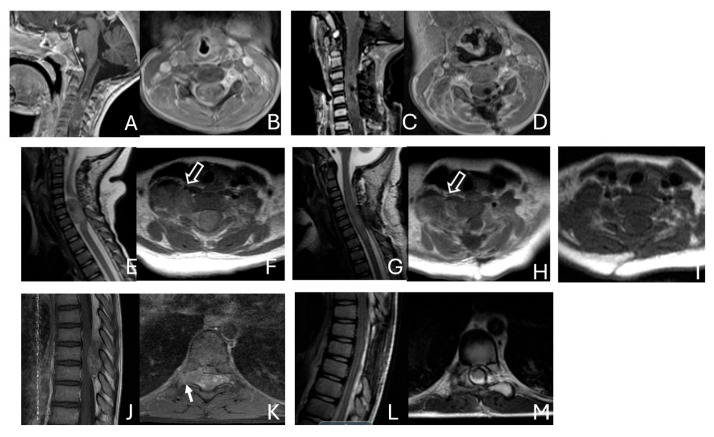
Spine ATRT. (**A**–**D**): MR images ((**A**), sagittal; (**B**), axial) of a 34-month-old girl show a heterogeneous intramedullary cervical tumor centered at C3–C5. Postoperative images ((**C**), sagittal; (**D**), axial) demonstrate gross-total resection. (**E**–**I**): T2-weighted sagittal MR (**E**) and post-contrast axial MR (**F**) of a 7-month-old girl show a C5–C7 intradural–extramedullary ATRT with extradural and extraspinal extension (open arrow). The intradural–extramedullary component from the C6 dorsal nerve root and the epidural component into the neural foramina were resected (**G**), sagittal; (**H**), axial through a laminoplastic laminotomy, note a persistent paraspinal tumor (open arrow). Following chemotherapy, the residual paraspinal component was subsequently resected via an anterior cervical approach ((**I**), axial). (**J**–**M**): Mid-thoracic MR images ((**J**), sagittal; (**K**), axial) of a 9-year-old boy show a dorsal epidural ATRT from T7 to T9 with right-sided neural foraminal extension (arrow). An epidural ATRT was removed through laminectomy ((**L**), sagittal; (**M**), axial).

**Figure 12 cancers-18-00008-f012:**
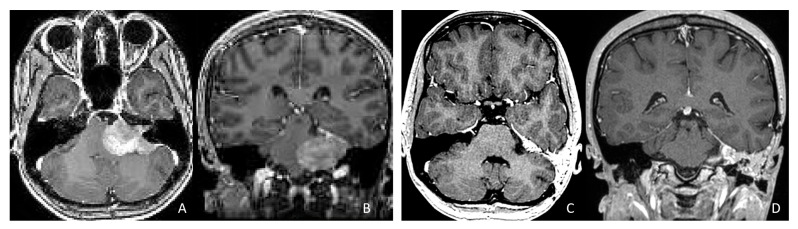
CPA ATRT. (**A**–**D**): MR images ((**A**): axial; (**B**): coronal) of a 20-year-old girl, the oldest in this cohort, who presented with left-sided hearing loss and partial facial weakness, reveal ATRT that appears to originate from the vestibular nerve. Surgical findings and post-resection MR images ((**C**): axial; (**D**): coronal) indicate no tumor invasion into the cerebellum or brainstem, which contrasts with typical presentations of infantile ATRT.

**Table 1 cancers-18-00008-t001:** Histogenesis of ATRT per subgroup.

	TYR	SHH			MYC
		1A	1B	2	
**Age**	young (12m)	young (8m)	older (107m)	young (13m)	older (27m)
**Location**	IT	ST	ST	IT/pineal	ST/IT/spine
**Origin**	MCP/ICV *	GE **	GE **	CAL at M-H ***	SC/GC ****
**Site**	CPA/cerebellum	basal ganglia/ventricle	basal ganglia/ventricle	SMV *****	nerves/cerebral cortex
**Linage**	neural crest	neural stem cell	neural stem cell	neural stem cell	neural crest
**Marker Gene**	melanosomal	SHH & NOTCH	SHH & NOTCH	SHH & NOTCH	MYC

Age: median age; IT: infratentorial, ST: supratentorial; * MCP/ICV; middle cerebellar peduncle/inferior cerebellar vermis; ** GE: ganglionic eminence; *** CAL at M-H: cerebellar anterior lobe at the midbrain-hindbrain boundary; **** SC/GC: Schwann cell/ganglion cell precursor; ***** SMV: superior medullary velum.

## Data Availability

The original contributions presented in this study are included in the article. Further inquiries can be directed to the corresponding author.

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
