# Peer review of "Histogenesis of Atypical Teratoid Rhabdoid Tumors: Anatomical and Embryological Perspectives"

_cancers, 2025, doi:10.3390/cancers18010008_

Round 1
Reviewer 1 Report
Comments and Suggestions for Authors
The author in this paper hypothesize that neural plate border progenitors with bipotent features represent a plausible intra-parenchymal cell of origin for ATRT. This hypothesis cannot be demonstrated due to the lack of transcriptomic and epigenetic data of these cases (because they have been studied retrospectively).
Author’s hypothesis is supported by an accurate anatomical topographic analysis of a large number of cases. Tumor epicenters and pattern of extension is very well documented pre, post operatively by imaging and by intraoperative neurosurgeon’s description. This analysis is done for each site more frequently involved in ATRT (Supratentorial, Infratentorial, Pineal, Spinal Cord) and it nicely show the peculiar intra and extra-axial and occasional extradural presentation of ATRT comparing with the other CNS tumors.
So because the type of accurate anatomical topographic analysis of a large cohort of cases , in parallel with an extensive and up-to-dated literature review, this paper deserves to be published; in fact it indicates clearly the distinct anatomical niches and age distribution for the entities described by molecular and methylation studies in ATRT.
I only suggest implementing the patients and methods paragraph clarifying better inclusion and exclusion criteria and age range for inclusion. Furthermore histopathological and immunoistochemical data of the 50 cases should be described more precisely.
I think that also a less generic description of the treatment delivered could be added.
Author Response
Thank you for reviewing my article and providing me valuable suggestions to improve its content.
Your suggestions are as follows, and I responded in the attached revised manuscript:
- “Clarifying better inclusion and exclusion criteria and age range for inclusion”.
Answer: it is addressed in “CLINICAL MATERIALS AND METHODS” highlighted in red in page 6-7
- “ histopathological and immunohistochemical data of the 50 cases should be described more precisely.”
Answer: Pertinent histological and IHC studies are described in “CLINICAL MATERIALS AND METHODS” highlighted in red as well in page 7
- “a less generic description of the treatment”
Answer: described in pages 17-18 highlighted in red in the section of “Postoperative managements”
Reviewer 2 Report
Comments and Suggestions for Authors
This is a comprehensive and informative review on ATRT and MRT.
However, the overall structure of the manuscript is somewhat ambiguous, as it mixes elements of a review article and a case series. I recommend shortening the case descriptions to improve focus and efficiency. There are too many MRI figures and their detailed legends; reducing the number of figures and emphasizing only key representative examples would enhance readability.
For the sections summarizing molecular pathogenesis and histogenesis, inclusion of a schematic or diagram would greatly help readers visualize the proposed (and complex) concepts.
Recent single-cell genomic studies provide the most up-to-date insights into the cell-of-origin question, but these are insufficiently discussed and should be incorporated.
In addition, the germline SMARCB1 mutations observed in adult schwannomatosis patients differ from those typically found in infantile ATRT; the relevant sentence on page 15 would benefit from further clarification.
Finally, the Conclusion section is rather long and contains redundant material. It could be made more concise to highlight the key messages of the review.
Author Response
Please see the revised manuscript. On the front page of the manuscript, the reviewer 2's comments and suggestions are described together with my responses and answers to them.
Reviewer 2
Thank you for reviewing my paper and providing valuable suggestions. Please read the revised manuscript and read my answers and responses to your questions and suggestions in the revised text highlighted in red.
Following are your suggestions and my responses:
- “Shortening the case descriptions to improve focus and efficiency. There are too many MRI figures and their detailed legends; reducing the number of figures and emphasizing only key representative examples would enhance readability.”
Answer: I have shortened the case descriptions and reduced the number of figures. However, further reductions are not advisable, as this may compromise the emphasis on the heterogeneous nature of ATRT, which is a key focus of this presentation. The postoperative imaging has to be retained to illustrate the tumor origins, providing readers with a better understanding.
- “Inclusion of a schematic or diagram would greatly help readers visualize the proposed (and complex) concepts:
Answer: I created Table 1 in Page 30.
- “Single-cell genomic studies provide the most up-to-date insights into the cell-of-origin question, but these are insufficiently discussed and should be incorporated”
Answer: I commented the value of single-cell genomic studies I the end of Conclusion in Page 32.
- “The germline SMARCB1 mutations observed in adult schwannomatosis patients differ from those typically found in infantile ATRT; the relevant sentence on page 15 would benefit from further clarification”
Answer: the difference between schwannomatosis and infantile ATRT is described in RTPS section of DISCUSSION in page 23
- “The Conclusion section is rather long and contains redundant material. It could be made more concise to highlight the key messages of the review.”
Answer: I have streamlined the conclusion section to eliminate redundancy while retaining all relevant information.